# The Role of Early Procalcitonin Determination in the Emergency Departiment in Adults Hospitalized with Fever

**DOI:** 10.3390/medicina57020179

**Published:** 2021-02-19

**Authors:** Marcello Covino, Antonella Gallo, Massimo Montalto, Giuseppe De Matteis, Maria Livia Burzo, Benedetta Simeoni, Rita Murri, Marcello Candelli, Veronica Ojetti, Francesco Franceschi

**Affiliations:** 1Emergency Medicine, Fondazione Policlinico Universitario A. Gemelli, IRCSS, 00168 Rome, Italy; bsimeoni@gmail.com (B.S.); marcello.candelli@policlinicogemelli.it (M.C.); vojetty@gmail.com (V.O.); francesco.franceschi@unicatt.it (F.F.); 2Faculty of Medicine and Surgery, Università Cattolica del Sacro Cuore, 00168 Rome, Italy; massimo.montalto@unicatt.it (M.M.); rita.murri@unicatt.it (R.M.); 3Department of Internal Medicine, Fondazione Policlinico Universitario A. Gemelli, IRCSS, 00168 Rome, Italy; antonella.gallo@policlinicogemelli.it (A.G.); dr.giuseppedematteis@gmail.com (G.D.M.); 4Emergency Department, Ospedale Generale M.G. Vannini, Istituto Figlie di San Camillo, 00177 Rome, Italy; maliburzo@gmail.com; 5Department of Infectious Diseases, Fondazione Policlinico Universitario A. Gemelli IRCCS, 00168 Rome, Italy

**Keywords:** procalcitonin, emergency department, qSOFA, sepsis, fever

## Abstract

*Background and Objectives:* Fever is one of the most common presenting complaints in the Emergency Department (ED). The role of serum procalcitonin (PCT) determination in the ED evaluation of adults presenting with fever is still debated. The aim of this study was to evaluate if, in adults presenting to the ED with fever and then hospitalized, the early PCT determination could improve prognosis. *Materials and Methods.* This is a retrospective, mono-centric study, conducted over a 10-year period (2009–2018). We analyzed consecutive patients ≥18 years admitted to ED with fever and then hospitalized. According to quick sequential organ failure assessment (qSOFA) at admission, we compared patients that had a PCT determination vs. controls. Primary endpoint was overall in-hospital mortality; secondary endpoints were in-hospital length of stay, and mortality in patients with bloodstream infection and acute respiratory infections. *Results.* The sample included 12,062 patients, median age was 71 years and 55.1% were men. In patients with qSOFA ≥ 2 overall mortality was significantly lower if they had a PCT-guided management in ED, (20.5% vs. 26.5%; *p* = 0.046). In the qSOFA < 2 group the mortality was not significantly different in PCT patients, except for those with a final diagnosis of bloodstream infection. *Conclusions.* Among adults hospitalized with fever, the PCT evaluation at ED admission was not associated with better outcomes, with the possible exception of patients affected by bloodstream infections. However, in febrile patients presenting to the ED with qSOFA ≥ 2, the early PCT evaluation could improve the overall in-hospital survival.

## 1. Introduction

Fever is one of the most common cause of Emergency Department (ED) access, accounting for 5% up to 15% of adult visits [1]. It represents an early warning sign of most infections, but could also be present in several non-infectious diseases, such as autoimmune diseases, and neoplasms [1].

An early antimicrobial administration demonstrated to be associated with reduced mortality in patients with bacterial infection and sepsis [1], thus the identification of fever of bacterial origin is essential for clinicians. Several clinical and laboratory tests were evaluated for this purpose, but none of them demonstrated an adequate sensitivity and specificity to definitively rule in a bacterial cause of fever [1].

Procalcitonin (PCT) is a precursor protein of calcitonin, expressed by human cells [2]. Its production is upregulated by pro-inflammatory cytokines like interleukin (IL)-1, IL-2, IL-6 and tumor necrosis factor alpha, and by bacterial endotoxins and lipopolysaccharide, while it is downregulated during viral infections [2,3]. Moreover, patients with autoimmune diseases or malignancies usually have low levels of PCT [4].

Several trials have reliably demonstrated the good performance of PCT in supporting the decision to start or stop antibiotic therapy in patients with suspected bacterial infections, leading to potential benefit in term of reduced length of in-hospital stays (LOS) and survival [5,6,7]. On these bases, in 2017 the United States Food and Drug Administration approved the use of PCT for guiding antibiotic therapy in acute respiratory infections and sepsis [8]. However, although PCT-guided management showed a good performance in selected populations, its role in patients with undifferentiated fever in ED is still unclear [9,10,11,12].

The aim of this study is to evaluate, in adults hospitalized with fever, if an early PCT determination at ED admission was associated to an improvement of the patient’s level outcomes, defined as the reduction of overall in-hospital mortality and LOS.

## 2. Materials and Methods

This is a retrospective study conducted in an academic medical center with an average attendance at the ED of about 75,000 patients annually (more than 87% adults). Based on electronic health records, we identified all consecutive patients admitted in ED for fever and then hospitalized during a 10-year period, between 1st January 2009 and 31 December 2018. We included in the analysis all patients with fever at presentation to ED or which reported fever within 24 h before ED access.

We excluded from our cohort patients with age <18 years, known HIV infection, acute leukemia or lymphoma, and patients in immunosuppressive treatment due to transplant.

### 2.1. Patients Characteristics and Clinical History

All the demographic and clinical variables were collected from hospital-based electronic health records. For each patient included in the analysis, we evaluated vital signs (systolic and diastolic blood pressure, heart rate, respiratory rate, peripheral oxygen saturation, body temperature) and clinical symptoms (including neurological impairment and acute respiratory failure) to assess the quick Sepsis Related Organ Failure Assessment (qSOFA) score [13] at ED admission.

Computerized clinical records were reviewed to acquire information about patient’s comorbidities, based on prior medical history and on the listing in the hospital discharge diagnosis. The comorbidities were used to assess the Charlson’s comorbidity score [14].

Chart review protocol is described in Appendix A.

### 2.2. Procalcitonin Sampling and Group Definition

The PCT was obtained in ED based on the clinical judgment of emergency physician at admission visit. Cutoff value of PCT serum level predictive of sepsis was set at 1 ng/mL, and a PCT interval between 0.5 and 1 ng/mL was considered as uncertain area. Procalcitonin determination was available 24 h a day.

All patients were categorized according to PCT determination. If PCT was assessed at ED presentation, patients were defined as “early PCT”; if PCT was not assessed at ED presentation, patients had a standard clinical-guided management and were defined as “controls”.

All patients with suspect infection and high PCT received empirical antibiotic therapy according to current guidelines. For patients in the “uncertain area” the antibiotic therapy was evaluated case by case. Local protocols for empirical therapy were stable during the study period.

### 2.3. Outcome Measures

The primary endpoint of the study was the all-cause in-hospital mortality.

As secondary endpoints, we evaluated the in-hospital mortality in the subgroups of patients with a discharge diagnosis of bloodstream infections, acute respiratory infections, and other site infections (cumulative).

Finally, we evaluated the overall length of hospital stay (LOS), calculated from the time of ED admission to the hospital discharge or death in PCT groups vs. controls.

Discharge diagnosis were ascertained by ICD code at hospital discharge.

### 2.4. Statistical Analysis and Chart Review Methodology

Six board certified emergency physicians reviewed the clinical records and inserted study variable in a digital database. Variables were determined according to a pre-definite patient’s form, based on the study protocol. To assess the intra-operator agreement of data extraction, 60 clinical records were randomly selected, and evaluated by all the six chart reviewers. We assessed the intra-operator reproducibility by Cohen’s kappa measured on categorical variables in the forms.

Categorical variables are presented as absolute numbers and percentages; continuous variables are presented as median [interquartile range]. Categorical variables were statistically compared by Chi-square test or Fisher exact test as appropriate. Continuous variables were compared by Mann Whitney U test.

We analyzed mortality rates and LOS in patients that had an early PCT in ED, compared to controls. Study endpoints were assessed separately in patients with qSOFA ≥ 2 and qSOFA < 2. *p* values were 2-sided, with a significance threshold of 0.05, and corrected in case of multiple groups comparison. Study analysis was conducted by SPSS version 25 (IBM, Armonk, NY, USA)

## 3. Results

### 3.1. Study Cohort

During the study period, 14,697 adult patients were hospitalized with a diagnosis of fever. Among these, 2635 patients were excluded, because not meeting inclusion criteria or for incomplete or inconsistent clinical records, yielding to a final study cohort of 12,062 patients. Chart review process demonstrated a good reproducibility in the randomly selected records, with a Cohen’s k 0.98 (95% CI 0.97–0.99). Enrollment details are reported in Appendix A. Baseline characteristics are shown in Table 1.

The serum PCT at ED access was determined in 3402 patients (28%), that represents the early PCT group. The remaining 8660 patients were used as control group.

A qSOFA < 2 was attributed to 11,136 (92.3%) patients; among them 3022 (27.1%) had a PCT determination in ED. A total of 826 patients (7.7%) had a qSOFA ≥ 2; 380 (46.0%) of them had a PCT determination in ED.

### 3.2. Early PCT Determination and In-Hospital Death

Overall, 1533 patients died (12.7%), and no differences were observed in death rates between patients in the early PCT group, compared to controls (Table 2).

In patients with qSOFA < 2, the early PCT determination in ED was not associated to a different mortality rate in the overall population. However, when considering the subgroup of patients with a final diagnosis of bloodstream infection the early PCT was associated to a significant better survival when compared to controls (21.0% vs. 30.6%; *p* = 0.003) (Table 2).

In patients with qSOFA ≥ 2, overall mortality was lower in patients which received a PCT assessment ED, being respectively 20.5% for early PCT group and 26.5% for controls (*p* = 0.046). When considering specific subgroups of infective diagnosis, the early PCT was generally associated to better survival rates, although not reaching statistical significance.

### 3.3. LOS and Early PCT Determination

Cumulative LOS of our patients was 10 (6–17) days. The early PCT patients had a significantly higher LOS compared to controls. This result was confirmed for both patients with qSOFA ≥ 2 and qSOFA < 2 (Table 3).

## 4. Discussion

The main finding of this study was that among adults admitted to ED with fever and subsequently hospitalized, an early PCT determination could improve prognosis in the group at higher risk of sepsis (qSOFA ≥ 2). Conversely, in patients with a low qSOFA score (<2), the early PCT determination was not associated to different outcomes, with the possible exception of patients affected by bloodstream infections.

The clinical management of patients with fever often represents a challenge for physicians, and determine whether fever is the expression of a harmful bacterial infection could be a challenging task in the ED setting. The available clinical and laboratory diagnostic tools could not be sufficient for an early diagnosis, and this particularly happens when the patient lacks the cognitive or physical ability to relay symptoms [15,16].

Direct identification of bacteria from blood culture and non-culture-based methodologies is expensive and time-consuming, and patients admitted to ED with fever are often exposed to an excess of broad-spectrum antibiotic therapy [17,18,19,20]. As a result, the interest on PCT to reduce both unnecessary and prolonged antibiotic therapy in these patients has grown in recent years.

In intensive care unit (ICU), the PCT-guided management of antibiotic therapy was associated to a mortality reduction [21]. At the same time, patients receiving PCT guided management had a shorter duration of antibiotic treatments [6,7,8,9,10,11]. This was confirmed by a meta-analysis on ICU patients with acute respiratory infections [8].

In study conducted in the ED setting, the early PCT was associated to a better discrimination of acute respiratory tract infections [16], and to a better prognosis in elderly patients with community-acquired pneumonia [22]. However, a multicenter randomized trial in patients admitted to ED with un-discriminate fever, showed that PCT testing did not reduce antibiotic prescription and 30-day mortality [12]. Similarly, patient benefit in term of mortality was not confirmed in the ED setting both for lower respiratory tract infections [23,24], and urinary tract infections [19,25].

As a whole, the PCT-guided management seems to have the most clinical benefit in high-risk populations while its utility in low-risk patients remains unclear [11]. Thus, conclusive evidence on the utility of early PCT determination in the ED is still lacking.

In our retrospective study, conducted in a large and heterogeneous population admitted to ED with fever, the early PCT showed a potential association to better survival in patients with qSOFA ≥ 2. This could likely be ascribed to an early start of antibiotic therapy or to a more aggressive clinical approach. These findings are in line with a recent meta-analysis confirming that in patients meeting sepsis-3 criteria, the PCT-guided management could be associated to an overall better survival [26]. However, analyzing the subgroups of patients with acute respiratory infections and bloodstream infections, our data demonstrate a slight reduction in overall mortality just in the latter ones, although not reaching the statistical significance. We can speculate that, in a population at high risk for sepsis (qSOFA ≥ 2), the overall effect of an early PCT management, although present, could be too low to be evidenced in a reduced sample size.

In patients with fever but at low risk of sepsis (qSOFA < 2), our data suggest that an extensive PCT determination in ED could have a limited influence on overall mortality. This is in line with a recent meta-analysis conducted on studies including septic and non-septic patients, in which the PCT-guided management did not show a significant benefit compared to standard clinical management [27].

Interestingly, our data demonstrated a significant reduction of mortality rate in the subgroup of patients accessing with qSOFA < 2 and having a discharge diagnosis of bloodstream infection. Several studies showed that PCT has a high diagnostic accuracy for bloodstream infection [28,29], although the false negative ratio is too high to use PCT alone to address this diagnosis [30]. Nevertheless, the association between PCT sampling and better survival in these patients could be due to an increased awareness for potential bacterial infection in these otherwise low-risk patients. Indeed, apart from the qSOFA score assessment, the clinical judgment of ED physicians should always play a key role in recognizing the most complex patients (i.e., patients with central venous catheter or other risk factors for bloodstream infection) [31]. In this setting, the role of PCT could be enhanced, increasing the confidence of ED physician for the need of an aggressive antibiotic therapy [31].

### Study Limitations

Although conducted on a large cohort of patients, some limitations are worth considering. First, this is a single center observational study, thus our result could not be generalizable to all EDs. Second, no established rule was defined to determine PCT assessment in ED, nor a specific PCT result management was operated. However, this latter limitation is diminished by the presence in our institution of an antibiotic stewardship team, which coordinate antibiotic prescriptions for every admitted patient. Finally, our observational study spans a decade, and the PCT sampling in ED considerably raised over the years (Appendix A).

## 5. Conclusions

Among adults admitted to ED with fever and hospitalized, those at high risk for sepsis (qSOFA ≥ 2) could have a better in-hospital survival if an early PCT determination is obtained in ED.

Conversely, in febrile patients with qSOFA < 2 at ED access, the early determination of PCT have a limited influence on overall prognosis, although in patients with high clinical suspicion of bloodstream infection it could be associated to improved outcomes if compared to standard clinical management.

As a result, a case-by-case analysis, and antibiotic stewardship are always recommended to maximize the clinical usefulness of the early PCT sampling for febrile adults in ED.

## Figures and Tables

**Table 1 medicina-57-00179-t001:** Demographic and clinical characteristics of the 12,062 patients included in the study.

Variable	Total Patients 12,062 pts	qSOFA < 2 11,136 pts	qSOFA ≥ 2 826 pts
Sex (Male)	6644 (55.1%)	6206 (55.2%)	438 (53.0)
Age (years)	71 (55–81)	70 (54–81)	78 (67–85)
Temperature °C	37.9 (37.0–38.8)	37.8 (36.9–38.7)	38.2 (37.5–39.1)
Heart Rate (b/min)	95 (80–110)	95 (80–110)	99 (83–114)
Systolic Blood Pressure (mmHg)	127 (110–145)	130 (114–145)	100 (90–110)
Diastolic Blood Pressure (mmHg)	70 (60–80)	71 (61–81)	60 (50–76)
Peripheral SaO_2_	95 (92–97)	95 (92–97)	94 (89–96)
Procalcitonin in Emergency Department	3402 (28.2%)	3022 (26.9%)	380 (46.0%)
Blood Culture in Emergency Department	2261 (18.7%)	1991 (17.7%)	270 (32.7%)
Charlson score ≥ 2	3244 (26.9%)	3010 (26.8%)	234 (28.3%)
Outcomes			
Infectious diagnosis (any) ^‡^	7437 (61.7)	6844 (60.9%)	593 (71.8%)
Acute respiratory inf. ^‡^	4525 (37.5)	4177 (37.2%)	348 (42.1%)
Bloodstream inf. ^‡^	919 (7.6%)	778 (6.9%)	141 (17.1%)
Other site infection ^‡^	3066 (25.4%)	2844 (25.3%)	222 (26.9%)
LOS ^#^ (days)	10 (6–17)	10 (6–17)	11 (7–18)
Deceased	1533 (12.7%)	1337 (11.9%)	196 (23.7%)

^‡^ Acute respiratory infections, Bloodstream infections, and other site infections were defined at hospital discharge; ^#^ Lenght of Hospital Stay; qSOFA: quick sequential organ failure assessment.

**Table 2 medicina-57-00179-t002:** In-hospital mortality rate in patients that an early procalcitonin (PCT) determination in emergency department (ED) vs. controls. Data are shown for all population and according to qSOFA at ED admission.

	Controls *n* 8860	Ealry PCT *n* 3402	*p* Value
All patients	1070/8660 (12.4%)	463/3402 (13.6%)	0.063
qSOFA < 2	Controls *n* 8214	Early PCT *n* 3022	*p* Value
All patients	952/8214 (11.6%)	385/3022 (12.7%)	0.095
Infectious diagnosis (any)	568/4831 (11.7%)	251/2013 (12.5%)	0.381
Acute respiratory inf.	459/3073 (14.9%)	174/1104 (15.8%)	0.512
Bloodstream infection	142/464 (30.6%)	66/248 (21.0%)	0.003
Other site infection	136/1974 (6.9%)	77/870 (8.9%)	0.067
qSOFA ≥ 2	Controls *n* 446	Early PCT *n* 380	*p* Value
All patients	118/446 (26.5%)	78/380 (20.5%)	0.046
Infectious diagnosis (any)	80/322 (24.8%)	55/271 (20.3%)	0.188
Acute respiratory inf.	66/218 (30.3%)	44/130 (33.8%)	0.488
Bloodstream infection	25/62 (40.3%)	25/79 (31.6%)	0.285
Other site infection	13/100 (13.0%)	10/122 (8.2%)	0.243

**Table 3 medicina-57-00179-t003:** Length of hospital stay (LOS) rate in patients that an early procalcitonin (PCT) determination in emergency department (ED) vs. controls. Data are shown for all population and according to qSOFA at ED admission.

		Controls *n* 8860	Early PCT *n* 3402	*p* Value
All Population		10 (6–17)	11 (7–18)	<0.001
qSOFA < 2		Controls *n* 8214	Early PCT *n* 3022	*p* Value
All patients		10 (6–17)	11 (7–18)	<0.001
Infectious disease diagnosis (any)		10 (6–17)	11 (7–18)	<0.001
qSOFA ≥ 2		Controls *n* 446	Ealry PCT *n* 380	*p* Value
All patients		10 (7–17)	11 (7–19)	0.044
Infectious disease diagnosis (any)		11 (7–17)	11 (7–19)	0.136

## Data Availability

The data presented in this study are available on reasonably request from the corresponding author.

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
