# Peer review of "The Role of Early Procalcitonin Determination in the Emergency Departiment in Adults Hospitalized with Fever"

_medicina, 2021, doi:10.3390/medicina57020179_

Round 1
Reviewer 1 Report
Thank you for giving me the opportunity too review this retrospective study on the role of PCT guided management of patients admitted with elevated temperature to the ED. The authors included 12,06 patients, created several subgroups based on qSOFA, and early PCT determination. They concluded that overall PCT guided management was not beneficial, but suggested that in patients with more severe presentation it could improve outcomes.
Comments
Using biomarkers to help clinical decision making in the ED has long history, and PCT is one of the most investigated parameter in this regard. It has been used to help diagnosing bacterial infection, to guide antibiotic therapy and it is also used as a prognostic marker. However, there are several open questions. Therefore, the aim of the authors to analyse the results of their practice is well justified. Unfortunately, the study is let down at several places, hence conclusions are difficult to drawn. These are my detailed comments:
- Inclusion criteria is very loose. On what basis was this decided?
- The terms “early PCT” vs “no early PCT” are confusing. Why don’t you call the groups simply PCT and Control groups?
- Do these cut offs mean that above 1 ng/mL patients received antimicrobials and below 0.5 they did not?
- What is the case regarding antibiotic administration in the “uncertain” range of 0.5-1?
- On what basis was PCT measurement decided? Personal preference of the attending physician or according to some sort of an institutional protocol?
- The study collected data from a 10-year period, during which the practice might have changed. How did the frequency of PCT determination changed over these years? In other words it would be nice to see if there was any difference in the distribution of patients in the PCT group vs Controls during the study period.
- The term “septic presentation” based on the qSOFA is also confusing. qSOFA is measure of severity based on organ dysfunction assessment, hence I would simply label these subgroups as >2 and <2 qSOFA.
- Please don’t repeat data depicted in tables in the text (age for example).
- How was infection diagnosed?
- The precision of PCT to diagnose infection should have been evaluated and possibly compared to other measures (temperature, etc).
- Was appropriateness of empirical antibiotic therapy evaluated? It should be as it has profound effect on outcomes.
- The incidence of blood stream infection is extremely high, the second most frequent finding in the “non septic” cohort and by far the highest in the septic group. Any explanation for that?
- I don’t really understand the message of Table 4. Why analyzing the deceased separately?
- There are too many subgroups in general, which makes the whole study quite complicated and gives the impression of some sort of a “data fishing” in search for significant results.
- The first part of the discussion is long and the statements are not put in context with the results of the study.
Reviewer 2 Report
It was a pleasure to review this manuscript. There are several areas where it is clear that the author(s) need help with English; I would suggest revision by an editor with better English language skills.
Further, the wording of the title, and through the paper, suggests that these were "ED patients" – but they were really patients who were seen in the ED and then hospitalized (line 69). Please do not frame this, then, as an ED-based study – it is a study that occurs in an ED/inpatient system. Yes, the PCT was obtained in the ED, but the patients were all managed in the hospital, and all of the endpoints were in-hospital based.
Title:
- The role of procalcitonin in managing pediatric fevers is under consideration, which was one reason that I was interested in reviewing this paper – but it turned out that this study excluded children. To avoid other readers being misled, I would suggest changing the word “patients” in the title to “adults.”
- Also, again, this study only included patients who were admitted to the ED and then admitted to the hospital. So, I would suggest that be reflected in the title as well. Maybe “admitted to the hospital after ED admission with fever?”
Abstract: often unclear.
- Again, state that this study only included adults.
- Reading the abstract alone suggests it looks at patients who are “admitted to ED” only – but the work included only inpatients!
- Some of the acronyms used are defined, but qSOFA is not.
- Also, I do not think “an indiscriminate use of PCT” is the right description, here. Maybe just “use”? Similar comment for line 250-251, although that entire sentence is incomprehensible to me.
Line 47 – The word “any” is actually the opposite of what you mean, I think. “None” perhaps?
Line 64 – what does the word “level” add to the phrase “level outcomes?” Why not just “outcomes”?
It is interesting that authors, and apparently the clinicians, consider a temperature over 37.5 C to be a fever. (line 72) I primarily care for children, but that temperature strikes me as rather low. Of interest, the US CDC uses 38 C to define fever in https://www.cdc.gov/quarantine/air/reporting-deaths-illness/definitions-symptoms-reportable-illnesses.html . Thus, I would consider either: making your definition of “fever” clear throughout the paper, beginning with the abstract, and justifying your definition with some relevant references; or re-doing the work with a more usual accepted definition of fever. Also, the authors included patients who did not have a fever in the ED, but also by report within the “first” (maybe they mean “most recent”? or “last’?) 24 hours before the ED visit (line 72). Was there any standardized way to assess recent temperatures? Did anti-pyretic administration matter?
Line 83-84: Grammar problems make this sentence about qSOFA hard to understand – is there literature to support this cutoff of 2 for sepsis/no sepsis? If so, please cite it here.
Line 91 – The fact that clinicians could choose to get a PCT or not is a major weakness of this retrospective study. It suggests that there will be a bias toward getting PCT when a doctor thinks a patient is sicker / more likely to be septic. This is mentioned in the limitations, but only briefly.
In any chart review, the Methods section needs to give more details of the review, such as who did the chart reviews, what training they had, how they resolved uncertainty or missing data, what sorts of quality checks were in place, etc. That is all absent here.
In the statistics section, please tell us if you did / not adjust for multiple comparisons. When I look at TABLE 4, I notice that there are a great number of comparisons, some of which may show statistically significant difference based on chance alone, without such adjustments.
Results: Although the p value for overall mortality difference was 0.063 > 0.05, this is so close to 0.05 that it is worth pointing-out – it is, in fact, likely there is actually a real difference in mortality between these two groups, with the “early PCT” group having higher mortality. This would confirm the bias of clinicians ordering PCT testing in sicker patients – those who died were probably likelier to have had a PCT drawn. Yet, early PCT seemed to be protective for those with qSOFA of 2 or more, which is a partial contradiction. Please discuss all of this.
Table 2 uses the word “in” to describe only three of the four patient groups.
3.1.1 – It is interesting to me that, in Table 2, the overall mortality difference for patients with qSOFA of 2 or more is statistically significant, but both of its two component subgroups do not have a statistically significant difference. Please tell us how that can be – is it due to small numbers in the subgroups? There is a similar finding in Table 3.
Line 163 – My manuscript did not have a Figure 1.
Discussion – line 215 – yes, it could be earlier antibiotics. Could it also be more liberal use of ICU-level care in patients with higher PCT, suggesting more supportive interventions?
Conclusions- please go back to your study aims, and make sure they are all addressed in the Conclusions. Your final sentence goes beyond your data – how is that justified? Conclusions in the text should match those in the Abstract.
References – I suspect Sager et al is reference #31 – is that right?
Round 2
Reviewer 1 Report
Thank you for your detailed answers for my comments. I believe that the manuscript has improved substantially. However, I still have some comments:
- I would change it: “The role of early PCT determination in the ED in patients hospitalized with fever”
- Abstract: I disagree with the term: “presenting to the ED for fever”. It doesn’t make sense. A patient is not admitted “for fever”, but “with fever”
- In general, the whole manuscript should be language edited by a native English speaker.
- Abstract (line 27): “…mortality in patients with specific discharge diagnosis.” What does it mean?
- Line 30: “PCT-guided assessment in ED”. Wouldn’t “PCT-guided management” be better?
- Regarding this sentence (line 264): “In our retrospective study, conducted in a large and heterogeneous real world population admitted to ED with fever…”. Please refrain from the quote “real world”. You were also referring to this couple of times in your answers to the reviewers’ comments as well. However, in a research paper, which states that there could be a mortality benefit of a specific strategy (ie: PCT-guided management), being “real world” is not a virtue. In fact, it is a severe limitation, that data were collected and patients managed in a “real world” manner and not according to a well-designed scientific protocol.
- I don’t understand this sentence (line 320): “Nevertheless, this consent us to better evaluate patients that in similar clinical context had a PCT sampling in ED vs. patients that had not.”
Reviewer 2 Report
Much improved; thank you for making the suggested changes, and others.
I still miss the information about chart review methods being in the paper, rather than in the supplemental materials. In the supplemental materials, the authors mention having done reliability testing - but we never see the results of it- was there good reliability? This should all be included in the Methods, and the results.
